# LATENT QUESTION REFORMULATION AND INFORMATION ACCUMULATION FOR MULTI-HOP MACHINE READING

## ABSTRACT

Multi-hop text-based question-answering is a current challenge in machine comprehension. This task requires to sequentially integrate facts from multiple passages to answer complex natural language questions. In this paper, we propose a novel architecture, called the Latent Question Reformulation Network (LQR-net), a multi-hop and parallel attentive network designed for question-answering tasks that require reasoning capabilities. LQR-net is composed of an association of **reading modules** and **reformulation modules**. The purpose of the reading module is to produce a question-aware representation of the document. From this document representation, the reformulation module extracts essential elements to calculate an updated representation of the question. This updated question is then passed to the following hop. We evaluate our architecture on the HOTPOTQA question-answering dataset designed to assess multi-hop reasoning capabilities. Our model achieves competitive results on the public leaderboard and outperforms the best current *published* models in terms of Exact Match (EM) and $F_1$ score. Finally, we show that an analysis of the sequential reformulations can provide interpretable reasoning paths.

## 1 INTRODUCTION

The ability to automatically extract relevant information from large text corpora remains a major challenge. Recently, the task of question-answering has been largely used as a proxy to evaluate the reading capabilities of neural architectures. Most of the current datasets for question-answering focus on the ability to read and extract information from a single piece of text, often composed of few sentences (Rajpurkar et al., 2016; Nguyen et al., 2016). This has strengthened the emergence of *easy questions* in the sense of Sugawara et al. (2018) and influenced the recent state-of-the-art models to be good at detecting patterns and named entities (Devlin et al., 2018; Yu et al., 2018; Wang et al., 2017). However they still lack actual reasoning capabilities.

The problem of reasoning requires machine comprehension models to *gather* and *compose* over different pieces of evidence spread across multiple paragraphs. In this work, we propose an original neural architecture that repeatedly reads from a set of paragraphs to aggregate and reformulate information. In addition to the sequential reading, our model is designed to collect pieces of information in parallel and to aggregate them in its last layer. Throughout the model, the important pieces of the document are highlighted by what we call a **reading module** and integrated into a representation of the question via our **reformulation module**. Our contributions can be summarised as follows:

- We propose a machine reading architecture, composed of multiple token-level attention modules, that collect information sequentially and in parallel across a document to answer a question,

- We propose to use an input-length invariant question representation updated via a dynamic max-pooling layer that compacts information form a variable-length text sequence into a fixed size matrix,

- We introduce an extractive reading-based attention mechanism that computes the attention vector from the output layer of a generic extractive machine reading model,

- We illustrate the advantages of our model on the HOTPOTQA dataset.

The remainder of the paper is organized as follows: Section 2 presents the multi-hop machine reading task, and analyses the required reasoning competencies. In Section 3, we detail our novel reading architecture and present its different building blocks. Section 4 presents the conducted experiments, several ablation studies, and qualitative analysis of the results. Finally, Section 5 discusses related work.

Our code to reproduce the results is publicly available at (*removed for review*).

## 2 TEXT-BASED QUESTION-ANSWERING AND MACHINE REASONING

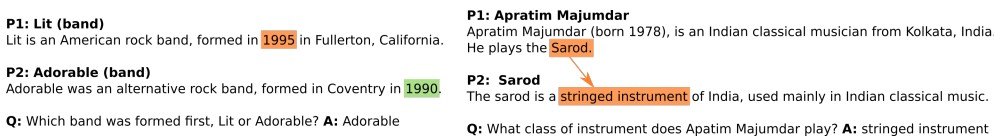

Figure 1: Examples of reasoning paths to answer two questions of the HOTPOTQA dataset. In this picture, we do not display the full paragraphs, but only the supporting facts.

The task of extractive machine reading can be summarized as follows: given a document $D$ and a question $Q$, the goal is to extract the span of the document that answers the question. In this work, we consider the explainable multi-hop reasoning task described in Yang et al. (2018) and its associated dataset: HOTPOTQA . We focus our experiments on the "**distractor**" configuration of the dataset. In this task, the input document $D$ is not a single paragraph but a set of ten paragraphs coming from different English Wikipedia articles. Answering each question requires gathering and integrating information from exactly two paragraphs; the eight others are distractors selected among the results of a tf-idf retriever (Chen et al., 2017). These required paragraphs are called the *gold* paragraphs. There are two types of questions proposed in this dataset: *extractive* ones where the answer is a span of text extracted from the document and binary *yes/no* questions. In addition to the answer, it is required to predict the sentences, also called *supporting facts*, that are necessary to produce the correct answer. This task can be decomposed in three subtasks: (1) categorize the answer among the three following classes: *yes, no, text span*, (2) if it is a span, predict the start and end positions of this span in the document, and (3) predict the supporting sentences required to answer the question. In addition to the "**distractor**" experiments, we show how our proposed approach can be used for open-domain question answering and evaluate the entire reading pipeline on the "**fullwiki**" configuration of the HotpotQA dataset. In this configuration, no supporting documents are provided, and it is required to answer the question from the entire Wikipedia corpus.

Among the competencies that multi-hop machine reading requires, we identify two major reasoning capabilities that human readers naturally exploit to answer these questions: sequential reasoning and parallel reasoning. **Sequential reasoning** requires reading a document, seeking a piece of information, then reformulating the question and finally extracting the correct answer. This is called multi-hop question-answering and refers to the *bridge* questions in HOTPOTQA . Another reasoning pattern is **parallel reasoning**, required to collect pieces of evidence for comparisons or question that required checking multiple properties in the documents. Figure 1 presents two examples from HOTPOTQA that illustrate such required competencies. We hypothesize that these two major reasoning patterns should condition the design of the proposed neural architectures to avoid restricting the model to one or the other reasoning skill.

## 3 THE MODEL

In this section, we describe the Latent Question Reformulation Network (LQR-net), shown in Figure 2. This multi-hop model is designed as an association of four modules: (1) an encoding module, (2) a reading module, (3) a question reformulation module, and (4) an answering module. (1) and (4) are input and output modules, whereas (2) and (3) constitute a hop, and are repeated respectively $T$ and $T-1$ times: the answering module does not require a last reformulation step.

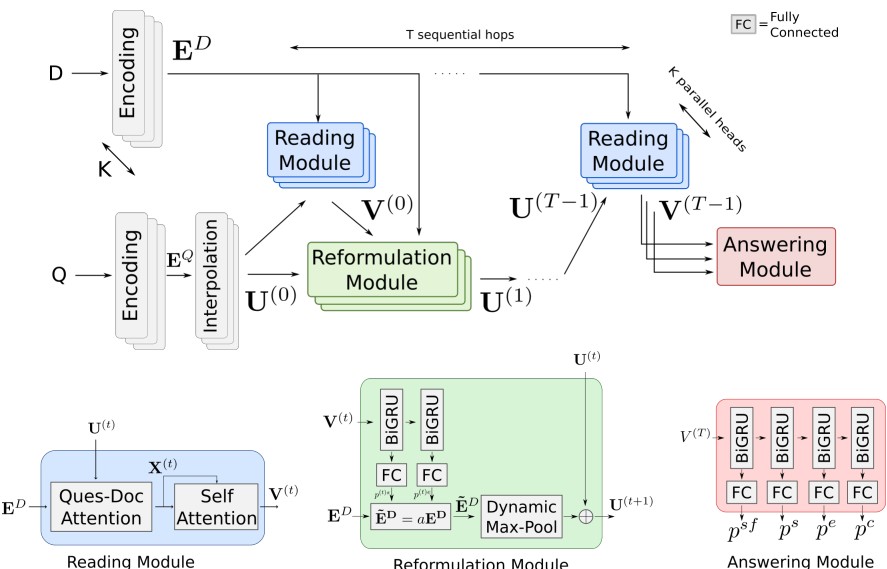

Figure 2: Overview of LQR-net with $K$ parallel heads and $T$ sequential reading modules. In this architecture, a latent representation of the question is sequentially updated to perform multi-hop reasoning. $K$ independent reading heads collect pieces of information before feeding them to the answering module. Sections 3 present the different building blocks of this end-to-end trainable model.

Given a document and a question, the reading module is in charge of computing a question-aware representation of the document. Then, the reformulation module extracts essential elements from this document representation and uses them to update a representation of the question in a latent space. This reformulated question is then passed to the following hop.

The model can have multiple heads, as in the Transformer architecture (Vaswani et al., 2017). In this case, the iterative mechanism is performed several times in parallel in order to compute a set of independent reformulations. The final representations of the document produced by the different heads are eventually aggregated before being fed to the answering module. This module predicts the answer and the *supporting facts* from the document. The following parts of this section describe each module that composes this model.

Note: The model is composed of $K$ independent reading heads that process the document and question in parallel. To not overload the notations of the next parts, we do not subscript all the matrices by the index of the head and focus on the description of one. The aggregation process of the multi-head outputs is explained in Section 3.5.

## 3.1 ENCODING MODULE

We adopt a standard representation of each token by using the pre-trained parametric language model BERT (Devlin et al., 2018). Let a document $D = \{p_1, p_2, \ldots, p_{10}\}$ be the set of input paragraphs, of respective lengths $\{n_1, \ldots, n_{10}\}$, associated to a question $Q$ of length $L$. These paragraphs are independently encoded through the pre-trained BERT model. Each token is represented by its associated BERT hidden state from the last layer of the model. The tokens representations are then concatenated to produce a global representation of the set of 10 paragraphs of total length $N = \sum_{i=1}^{10} n_i$. The representations are further passed through a Bidirectional Gated Recurrent Unit (BiGRU) (Cho et al., 2014) to produce the final representation of the document $\boldsymbol{E}^D \in \mathbb{R}^{N \times 2h}$ and question $\boldsymbol{E}^Q \in \mathbb{R}^{L \times 2h}$, where $h$ is the hidden state dimension of the BiGRUs.

$$\boldsymbol{E}^Q = \text{BiGRU}(\text{BERT}(Q)), \qquad \boldsymbol{E}^D = \text{BiGRU}([\text{BERT}(p_1); \ldots; \text{BERT}(p_{10})]), \qquad (1)$$

where $[;]$ is the concatenation operation.

To compute the first representation of the question $\boldsymbol{U}^{(0)}$, we use an interpolation layer to map $\boldsymbol{E}^Q \in \mathbb{R}^{L \times 2h}$ to $\boldsymbol{U}^{(0)} \in \mathbb{R}^{M \times 2h}$ where $M$ is an hyperparameter of the model. Intuitively, $\mathbb{R}^{M \times 2h}$ corresponds to the space allocated to store the representation of the question and its further reformulations. It does not depend on the length of the original question $L$.

## 3.2 READING MODULE

Our model is composed of $T$ hops of *reading* that sequentially extract relevant information from a document regarding the current reformulation of the question. At step $t$, given a representation of the reformulated question $\boldsymbol{U}^{(t)} \in \mathbb{R}^{M \times 2h}$ and a representation of the document $\boldsymbol{E}^D \in \mathbb{R}^{N \times 2h}$, this module computes a question-aware representation of the document. This module is a combination of two layers: a document-question attention followed by a document self-attention.

**Document-Question Attention:** We first construct the interaction matrix between the document and the current reformulation of the question $\boldsymbol{S} \in \mathbb{R}^{N \times M}$ as:

$$S_{i,j} = \boldsymbol{w}_1 \boldsymbol{E}_{i,:}^D + \boldsymbol{w}_2 \boldsymbol{U}_{j,:}^{(t)} + \boldsymbol{w}_3(\boldsymbol{E}_{i,:}^D \odot \boldsymbol{U}_{j,:}^{(t)}), \tag{2}$$

where $\boldsymbol{w}_1, \boldsymbol{w}_2, \boldsymbol{w}_3$ are trainable vectors of $\mathbb{R}^{2h}$ and $\odot$ the element-wise multiplication. Then, we compute the document-to-question attention $\boldsymbol{C}^q \in \mathbb{R}^{N \times 2h}$ :

$$P_{i,j} = \frac{\exp(S_{i,j})}{\sum_{k=1}^M \exp(S_{i,k})}, \quad \boldsymbol{C}_{i,:}^q = \sum_{j=1}^M P_{i,j} \boldsymbol{U}_{j,:}^{(t)}. \tag{3}$$

And the question-to-document attention $\boldsymbol{q}^c \in \mathbb{R}^{2h}$:

$$m_i = \max_{j \in \{1,\dots,M\}} S_{i,j}, \quad \boldsymbol{p} = \text{softmax}(\boldsymbol{m}), \quad \boldsymbol{q}^c = \sum_{j=1}^N p_j \boldsymbol{E}_{j,:}^D. \tag{4}$$

Finally, we compute the question-aware representation of the document $\boldsymbol{X}^{(t)} \in \mathcal{R}^{N \times 8h}$:

$$\boldsymbol{X}_{i,:}^{(t)} = [\boldsymbol{E}_{i,:}^D; \boldsymbol{C}_{i,:}^q; \boldsymbol{E}_{i,:}^D \odot \boldsymbol{C}_{i,:}^q; \boldsymbol{q}^c \odot \boldsymbol{C}_{i,:}^q], \tag{5}$$

where $[;]$ concatenation operation. Finally, we use a last BiGRU that reduces the dimension of $\boldsymbol{X}^{(t)}$ to $N \times 2h$. This specific attention mechanism was first introduced in the Bidirectional Attention Flow model of Seo et al. (2017). We hypothesize that such token-level attention will produce a finer-grained representation of the document compared to sentence-level attention used in state-of-the-art Memory Network architectures.

**Document Self-Attention:** So far, the contextualization between the ten paragraphs has only be done by the BiGRUs of equation 1. One limitation of the current representation of the document is that each token has very limited knowledge of the other elements of the context. To deal with long-range dependencies, we apply this same attention mechanism between the question-aware representation of the document, $\boldsymbol{X}^{(t)}$, and itself to produce the reading module output $\boldsymbol{V} \in \mathbb{R}^{N \times 2h}$. This self-contextualization of the document has been found useful in our experiments as presented in the ablation analysis of Section 4.3.

## 3.3 QUESTION REFORMULATION MODULE

A reformulation module $t$ takes as input the output of the previous attention module $\boldsymbol{V}^{(t)}$, the previous representation of the reformulated question $\boldsymbol{U}^{(t)}$, and an encoding of the document $\boldsymbol{E}^D$. It produces an updated reformulation of the question $\boldsymbol{U}^{(t+1)}$.

**Reading-based Attention:** Given $\boldsymbol{V}^{(t)}$ we compute $\boldsymbol{p}^{(t)s} \in \mathbb{R}^N$ and $\boldsymbol{p}^{(t)e} \in \mathbb{R}^N$ using two BiGRUs followed by a linear layer and a softmax operator. They are computed from:

$$\begin{aligned} \boldsymbol{Y}^{(t)s} &= \text{BiGRU}(\boldsymbol{V}^{(t)}) & \boldsymbol{Y}^{(t)e} &= \text{BiGRU}(\boldsymbol{Y}^{(t)s}) \\ \boldsymbol{p}^{(t)s} &= \text{softmax}(\boldsymbol{w}_s \boldsymbol{Y}^{(t)s}) & \boldsymbol{p}^{(t)e} &= \text{softmax}(\boldsymbol{w}_e \boldsymbol{Y}^{(t)e}), \end{aligned} \tag{6}$$

where $\boldsymbol{w}_e$ and $\boldsymbol{w}_s$ are trainable vectors of $\mathbb{R}^h$. The two probability vectors $\boldsymbol{p}^{(t)s}$ and $\boldsymbol{p}^{(t)e}$ are not used to predict an answer but to compute a reading-based attention vector $\boldsymbol{a}^{(t)}$ over the document. Intuitively, these probabilities represent the belief of the model at step $t$ of the probability for each word to be the beginning and the end of the answer span. We define the reading-based attention of a token as the probability that the predicted span has started before this token and will end after. It can be computed as follows:

$$a_i^{(t)} = \Big( \sum_{k=0}^{i} p_k^{(t)s} \Big) \Big( \sum_{k=i}^{N} p_k^{(t)e} \Big). \tag{7}$$

Finally, we use these attention values to re-weight each token of the document representation. We compute $\tilde{E}^{(t)D} \in \mathcal{R}^{N \times 2h}$ with:

$$\tilde{E}_{i,j}^{(t)D} = a_j^{(t)} E_{i,j}^D. \tag{8}$$

**Dynamic Max-Pooling:** This layer aims at collecting the relevant elements of $\tilde{\boldsymbol{E}}^{(t)D}$ to add to the current representation of dimension $M \times 2h$. We partition the row of the initial sequence into $M$ approximately equal parts. It produces a grid of $M \times 2h$ in which we apply a max-pooling operator in each individual window. As a result, a matrix of fixed dimension adequately represents the input, preserving the global structure of the document, and focusing on the important elements of each region. This can be seen as an adaptation of the dynamic pooling layer proposed by Socher et al. (2011).

Formally, let $\tilde{\boldsymbol{E}}^{(t)D}$ be the input matrix representation, we dynamically compute the kernel size, $w$, of the max-pooling according to the length of the input sequence and the required output shape: $w = \lceil \frac{N}{M} \rceil$, $\lceil \cdot \rceil$ being the ceiling function. Then the output representation of this pooling layer will be $\boldsymbol{O}^{(t)} \in \mathbb{R}^{M \times 2h}$ where

$$O_{i,j}^{(t)} = \max_{k \in \{iw, \dots, (i+1)w\}} (S_{k,j}). \tag{9}$$

Finally, to compute the updated representation of the question $\boldsymbol{U}^{(t+1)} \in \mathbb{R}^{M \times 2h}$, we sum $\boldsymbol{U}^{(t)}$ and $\boldsymbol{O}^{(t)}$.

## 3.4 ANSWERING MODULE

The answering module is a sequence of four BiGRUs, each of them followed by a fully connected layer. Their respective goal is to supervise (1) the supporting facts $p^{\text{sf}}$, (2) the answer starting and (3) ending probabilities, $\boldsymbol{p}^e$, $\boldsymbol{p}^s$, of each word of the document. (4) The last layer is used as a three-way classifier to predict $p^c$ the probability of the answer be classified as *yes*, *no* or a *span of text*.

$$\boldsymbol{Y}^{\text{sf}} = \text{BiGRU}(\boldsymbol{V}^{(t)}) \quad \begin{aligned} \boldsymbol{Y}^s &= \text{BiGRU}(\boldsymbol{Y}^{\text{sf}}) \\ \boldsymbol{p}^s &= \text{softmax}(\boldsymbol{w}_s \boldsymbol{Y}^s) \end{aligned} \quad \begin{aligned} \boldsymbol{Y}^e &= \text{BiGRU}(\boldsymbol{Y}^s) \\ \boldsymbol{p}^e &= \text{softmax}(\boldsymbol{w}_e \boldsymbol{Y}^e) \end{aligned} \quad \begin{aligned} \boldsymbol{Y}^c &= \text{BiGRU}(\boldsymbol{Y}^e) \\ \boldsymbol{p}^c &= \text{softmax}(\boldsymbol{w}_c \boldsymbol{Y}^c) \end{aligned} \tag{10}$$

where $\boldsymbol{w}_s \in \mathbb{R}^h$, $\boldsymbol{w}_e \in \mathbb{R}^h$, $\boldsymbol{W}_c \in \mathbb{R}^{h \times 3}$ are trainable parameters.

To predict the supporting facts, we construct a sentence based representation of the document. Each sentence is represented by the concatenation of its starting and ending supporting fact tokens from $\boldsymbol{Y}^{\text{sf}}$. We compute $p_{i,j}^{\text{sf}}$ the probability of sentence $j$ of example $i$ of being a supporting fact with a linear layer followed by a sigmoid function.

## 3.5 MULTI-HEAD VERSION

We define a multi-head version of the model. In this configuration, we use a set of independent parallel heads. All heads are composed of the same number of reading and reformulation modules. Each head produces a representation $\boldsymbol{V}_k^{(T)}$ of the document. We finally sum these $K$ matrices to compute the input of the answering block.

### 3.6 TRAINING

We jointly optimize the model on the three subtasks (supporting facts, span position, classifier *yes/no/span*) by minimising a linear combination of the supporting facts loss $\mathcal{L}_{\text{sf}}$, the span loss $\mathcal{L}_{\text{span}}$ and the class loss $\mathcal{L}_{\text{class}}$. Let $N_d$ be the number of examples in the training dataset. $\mathcal{L}_{\text{sf}}(\theta)$ is defined by:

$$\mathcal{L}_{\text{sf}}(\theta) = \frac{1}{N_d} \sum_i^{N_d} \frac{1}{\text{nbs}_i} \sum_j^{\text{nbs}_i} (p_{i,j}^{\text{sf}} - y_{i,j}^{(1)})^2, \tag{11}$$

where $\text{nbs}_i$ corresponds to the number of sentences in the document $i$. $y_{i,j}^{(1)}$ being 1 if the sentence $j$ of the document $i$ is a supporting fact otherwise 0.

Selecting the answer in multi-hop reading datasets is a weakly supervised task. Indeed, similarly to the observations of Min et al. (2019a) for open-domain question-answering and discrete reasoning tasks, it is frequent for a given answer of HOTPOTQA to appear multiple times in its associated document. In our case, we assume that all the mentions of the answer in the supporting facts are related to the question. We tag as a valid solution, the start and end positions of all occurrences of the answer in the given supporting facts.

$\mathcal{L}_{\text{span}}(\theta)$ is defined by:

$$\mathcal{L}_{\text{span}}(\theta) = \frac{1}{N_d} \sum_i^{N_d} \frac{1}{2} D_{\text{KL}}(p_i^s \| y_i^{(2)}) + D_{\text{KL}}(p_i^e \| y_i^{(3)}) \tag{12}$$

where $y_i^{(2)} \in \mathbb{R}^N, y_i^{(3)} \in \mathbb{R}^N$ are vectors containing the value $1/n_i$ at the start, end positions of all the occurrences of the answer, 0 otherwise; $n_i$ being the number of occurrences of the answer in the context.

$\mathcal{L}_{\text{class}}(\theta)$ is defined by:

$$\mathcal{L}_{\text{class}}(\theta) = -\frac{1}{N_d} \sum_i^{N_d} \log(p_{i,y_i^{(4)}}^c), \tag{13}$$

where $y_i^{(4)}$ corresponds to the index of the label of the question type {*yes*, *no*, *span*}. We finally define the training loss as follows:

$$\mathcal{L}(\theta) = \mathcal{L}_{\text{class}}(\theta) + \alpha \mathcal{L}_{\text{span}}(\theta) + \beta \mathcal{L}_{\text{sp}}(\theta), \tag{14}$$

where $\alpha$ and $\beta$ are hyperparameters tuned by cross-validation.

## 4 EXPERIMENTS

### 4.1 DATA AUGMENTATION

In the original HOTPOTQA dataset, the two *gold* paragraphs required to answer a given question come with eight distractor paragraphs. These eight distractor paragraphs, collected from Wikipedia, are selected among the results of a bigram tf-idf retriever (Chen et al., 2017) using the question as the query. As an augmentation strategy, we created additional "easier" examples by combining the two *gold* paragraphs with eight other paragraphs randomly selected in the dataset. For each example of the original training set, we generate an additional "easier" example. These examples are shuffled in the dataset.

### 4.2 IMPLEMENTATION DETAILS

Our model is composed of 3 parallel heads ($K = 3$) each of them composed of two reading modules and one reformulation module ($T = 2$). We set the hidden dimension of all the GRUs to $d = 80$. We use $M = 100$ to allocate a space of $\mathbb{R}^{100 \times 160}$ to store the question and its reformulations. We use

| Model | Answer | | Sup Fact | | Joint | |
|---|---|---|---|---|---|---|
| | EM | $F_1$ | EM | $F_1$ | EM | $F_1$ |
| LQR-net (our) | **60.20** | **73.78** | 56.21 | 84.09 | **36.56** | **63.68** |
| DFGN (Qiu et al., 2019) | 56.31 | 69.69 | 51.50 | 81.62 | 33.62 | 59.82 |
| QFE (Nishida et al., 2019) | 53.86 | 68.06 | **57.75** | **84.49** | 34.63 | 59.61 |
| Baseline Model (Yang et al., 2018) | 45.60 | 59.02 | 20.32 | 64.49 | 10.83 | 40.16 |
| DecompRC (Min et al., 2019b) | 55.20 | 69.63 | N/A | N/A | N/A | N/A |
| Self-Assembling NMN (Jiang & Bansal, 2019) | 49.58 | 62.71 | N/A | N/A | N/A | N/A |

Table 1: Performance comparison on the private test set of HOTPOTQA in the distractor setting. We compare our model, in term of Exact Match and $F_1$ scores, against the *published* models at the time of submission (September 25th). Our submission is tagged as *LQR-net 2 + BERT-Base (single model)* on the official leaderboard (`https://hotpotqa.github.io/`).

pre-trained *BERT-base-cased* model (Devlin et al., 2018) and adapt the implementation of *Hugging Face*[1] to compute embedding representations of documents and questions. We optimize the network using the Adam optimizer (Kingma & Ba, 2015) with an initial learning rate of $1e^{-4}$. We set $\alpha$ to 1 and $\beta$ to 10. All these parameters have been defined through cross-validation.

## 4.3 RESULTS AND ABLATION ANALYSIS

Table 1 presents the performance of our LQR-net on the distractor setting of the HOTPOTQA dataset. We compare our model against the *published* approaches evaluated on the HOTPOTQA dataset. We can see from this table that our model achieves strong performance on the answer prediction task. It outperforms the current best model by 3.9 points of EM and 4.1 points of $F_1$ score. Our model also achieves competitive performance for the evidence extraction task. The LQR-net achieves state-of-the-art performance on the joint task improving the best *published* approaches by 2.9 points on EM and 3.9 points of $F_1$.

| Model | Answer | | Sup Fact | | Joint | |
|---|---|---|---|---|---|---|
| | EM | $F_1$ | EM | $F_1$ | EM | $F_1$ |
| LQR | **60.0** | **74.1** | **55.8** | 83.9 | **36.5** | **64.0** |
| - Data aug | 59.3 | 73.4 | 52.8 | **84.2** | 34.4 | 63.6 |
| CE Loss | 59.6 | 73.6 | 52.7 | 83.5 | 34.4 | 63.2 |
| K = 1 | 59.2 | 73.2 | 48.9 | 83.8 | 31.6 | 63.0 |
| - Self-Att | 53.4 | 66.8 | 48.9 | 79.2 | 30.1 | 55.7 |
| T = 1 | 53.4 | 67.2 | 48.3 | 78.2 | 28.8 | 55.1 |
| M = 1 | 51.8 | 65.2 | 42.1 | 72.1 | 25.8 | 50.7 |

Table 2: Comparison of different architectures and model choices against the best configuration on the development set of HotpotQA.

To evaluate the impact of the different components of our model, we perform an ablation analysis. Table 2 presents the results of this analysis.

**Impact of sequential and parallel reading:** We study the contributions of the sequentiality in the model and of the multiple parallel heads. We compare our model to a similar architecture without the sequential reformulation ($T = 1$). We find that this sequential association of reading modules and reformulation modules is a critical component. $F_1$ score decreases by 6.9 points for the answer prediction task and 5.7 points for the evidence extraction task when the model does not have the capability to reformulate the question.

---

[1]`https://github.com/huggingface/pytorch-transformers`

The impact of the parallel heads is more limited than the sequentiality but still remains significant. Indeed, the configuration that uses only a single head ($K = 1$) stands 1 $F_1$ points below the best model on the joint metric.

**Weak supervision of the answer:** In this work, we propose to label as positive all occurrences of the answer in the supporting facts. We compare this configuration to the standard approach, where only the first occurrence of the answer is labeled as positive and the others as negative. In this last configuration, the span loss corresponds to a cross-entropy loss (CE loss) between the predicted start and end probabilities and the target positions. This decreases the joint $F_1$ score by $0.8$ points.

**Impact of the self-attention layer:** We study the impact of the self-attention layer in the reading module. We found that this self-attention layer is an essential component in the reading process. Indeed, when we omit this layer, the $F_1$ score decreases by $8.3$ points on the joint metric. This outlines the necessity to be able to propagate long-range information between the different paragraphs and not only in the local neighborhood of a token. Compared to previously proposed approaches, this layer does not rely on any handcrafted relationship across words.

**Question as a single vector:** Finally, we study the case where the question representation is reduced to a vector of $\mathbb{R}^{2h}$ ($M = 1$). This configuration achieves the worst results of our analysis, dropping the joint $F_1$ score by $13.3$ points and highlights the importance of preserving a representation of the question as a matrix to maintain its meaning.

## 4.4 OPEN-DOMAIN EXPERIMENTS

In this part, we describe how we integrated our model into an entire reading pipeline for open-domain question answering. In this setting, no supporting documents are associated to each question, and it is required to retrieve relevant context from large text corpora such as Wikipedia. We adopt a two-stage process, similar to Chen et al. (2017); Clark & Gardner (2018), to answer multi-hop complex questions based on the 5 million documents of Wikipedia. First, we use a paragraph retriever to select a limited amount of relevant paragraphs from a Wikipedia dump, regarding a natural language question. Second, we fed our LQR model with the retrieved paragraphs to extract the predicted answer. We evaluate this approach on the open-domain configuration of the HotpotQA dataset called ***fullwiki***.

We use a standard TF-IDF based paragraph retriever to retrieve the paragraphs the most related to the question. In addition to these paragraphs, we consider as relevant their *neighbors* in the Wikipedia graph, i.e. the documents linked to them by hyperlinks. In our experiments, we considered as relevant, the top 10 paragraphs and their associated *neighbors*.

Table 3 shows the results of our approach compared to other published models. Although we are using a very simple retriever, only based on TF-IDF, we report strong results on the open-domain question answering task of HotpotQA. The only published approach (Nie et al., 2019) that outperforms us being a combination of sentence/paragraph retrieval based on BERT encodings.

## 4.5 QUALITATIVE ANALYSIS

**Question Reformulation and Reasoning Chains:** Because our model reformulates the question in a latent space, we cannot directly visualize the text of the reformulated question. However, one way to assess the effectiveness of this reformulation is to analyze the evolution of $p^s$ and $p^e$ across the two hops of the model. We present in Figure 3 an analysis of the evolution of these probabilities on two *bridge* samples of the development dataset. We display the reading-based attention, that corresponds to the probabilities for each word to be part of the predicted span, computed from $p^s$ and $p^e$ in Equation 7. These examples show this attention before the first reformulation of the question and in the answering module.

From these observations, we can see that the model tends to follow a natural reasoning path to answer *bridge* questions. Indeed, before the first reformulation module, the attentions tend to focus on the first step of reasoning. For the question *"What award did the writer of Never Let Me Go novel win in 1989?"*, the model tends to focus on the name of the writer at the first step, before jumping the award description in the second step. Similarly, for the question *"What is the population according to the 2007 population census of the city in which the National Archives and Library of Ethiopia is*

| Model | Answer | | Sup Fact | | Joint | |
|---|---|---|---|---|---|---|
| | EM | $F_1$ | EM | $F_1$ | EM | $F_1$ |
| SemanticRetrievalMRS (Nie et al., 2019) | **46.50** | **58.80** | **39.90** | **71.5** | **26.6** | **49.2** |
| LQR-net (our) | 43.00 | 54.0 | 30.10 | 58.90 | 18.90 | 39.20 |
| GOLDEN Retriever[†] (Qi et al., 2019) | 37.92 | 48.58 | 30.69 | 64.4 | 18.04 | 39.13 |
| CogQA (Ding et al., 2019) | 37.60 | 49.40 | 23.10 | 58.5 | 12.2 | 35.3 |
| MUPPET (Feldman & El-Yaniv, 2019) | 31.07 | 40.42 | 17.00 | 47.71 | 11.76 | 27.62 |
| QFE[†] (Nishida et al., 2019) | 28.70 | 38.10 | 14.20 | 44.40 | 8.69 | 23.1 |
| Baseline Model (Yang et al., 2018) | 24.68 | 34.36 | 5.28 | 40.98 | 2.54 | 17.73 |
| DecompRC (Min et al., 2019b) | N/A | 43.26 | N/A | N/A | N/A | N/A |

Table 3: Performance comparison on the development set of HOTPOTQA in the *fullwiki* setting. We compare our model in terms of Exact Match and $F_1$ scores against the *published* models at the time of submission (November 15th). † indicates that the paper does not report the results on the development set of the dataset; we display their results on the test set.

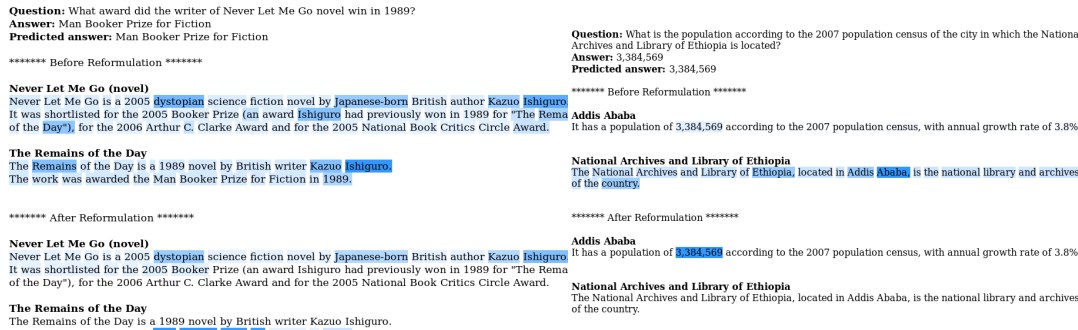

Figure 3: Distribution of the probabilities for each word to be part of the predicted span, before the first reformulation module and in the answering module. We display the reading-based attention computed in Equation 7 and the reading-based attention computed from $p^s$ and $p^e$ from Equation 10. In these examples, we show only the supporting facts.

*located?"* we can see the model focusing on Addis Ababa at the first step, i.e the name of the city where the National Archives and Library of Ethiopia are located and then jumping to the population of this city in the next hop. We display more visualizations of the sequential evolution of the answer probabilities in Appendix A.

**Limitations:** We manually examine one hundred errors produced by our multi-step reading architecture on the development set of HOTPOTQA . We identify three recurrent cases of model failure: (1) the model stops at the first hop of required reasoning, (2) the model fails at comparing two properties, and (3) the answer does not match all the requirements of the question. We illustrate these three recurrent types of error with examples from the dataset in Appendix B.

During this analysis of errors, we found that in only 3% of the cases, the answer is selected among one of the distractor paragraphs instead of a *gold* one. Our architecture successfully detects the relevant paragraphs regarding a question even among similar documents coming from a tf-idf retriever. Moreover, there are no errors where the model produces a binary *yes/no* answer instead of extracting a text span and vice versa. Identifying the type of question is not challenging for the model. This might be explained by the question's "patterns" that are generally different between binary *yes/no* and extractive questions.

## 5 RELATED WORK

**Multi-hop Machine Comprehension:** The question-answering task has recently increased its popularity as a way to assess machine reading comprehension capabilities. The emergence of large

scale datasets such as CNN/Daily Mail, (Hermann et al., 2015), SQuAD (Rajpurkar et al., 2016) or MSMARCO (Nguyen et al., 2016) have encouraged the development of multiple machine reading models (Devlin et al., 2018; Wang et al., 2018; Tan et al., 2017). These models are mainly composed of multiple attention layers that update the representation of the document conditioned by a representation of the question.

However, most of this work focuses on the ability to answer questions from a single paragraph, often limited to a few sentences. Weston et al. (2015a); Joshi et al. (2017) were the first attempts to introduce the task of multi-documents question-answering. QAngaroo (Welbl et al., 2018) is another dataset designed to evaluate multi-hop reading architectures. However, state-of-the-art architectures on this task (Zhong et al., 2019; Cao et al., 2019) tend to exploit the structure of the dataset by using the proposed candidate spans as an input of the model.

Recently, different approaches have been developed for HOTPOTQA focusing on the multiple challenges of the dataset. Nishida et al. (2019) focuses on the evidence extraction task and highlight its similarity with the extractive summarization task. Related works also focus on the interpretation of the reasoning chain with an explicit decomposition of the question (Min et al., 2019b) or a decomposition of the reasoning steps (Jiang & Bansal, 2019). Other models like Qiu et al. (2019) aim at integrating a graph reasoning type of attention where the nodes are recognized by a BERT NER model over the document. Moreover, this model leverages on handcrafted relationships between tokens.

Related to our approach, different papers have investigated the idea of question reformulation to build multi-hop open-domain question answering models. Das et al. (2019) proposes a framework composed of iterative interaction between a document retriever and a reading model. The question reformulation is performed by a *multi-step-reasoner* module trained via reinforcement learning. Similarly, Feldman & El-Yaniv (2019) introduces a multi-hop paragraph retriever. They propose a reformulation component integrated into a retrieving pipeline to iteratively retrieve relevant documents. These works are complementary to ours by focusing mostly on the document retrieving part of the problem while we focus on the answer extraction task, and could be combined together.

**Memory Networks**: Memory networks are a generic type of architecture Weston et al. (2015b); Sukhbaatar et al. (2015); Miller et al. (2016) designed to iteratively collect information from memory cells using attention mechanism. They have been used to read from sentences, paragraphs, and knowledge bases. In these models, the answer layer uses the last value of the controller to predict the answer. Two main differences with our architecture are the representation of the controller and the associated attention mechanism. Indeed, in these models, the controller is reduced to a single vector, and the attention mechanism is based on a simple dot-product between each token of the document and the representation of the controller. We utilize a token-level attention mechanism compared to the sentence-level one, classically used in Memory Networks.

**Transformer Networks:** The transformer architecture has been introduced by Vaswani et al. (2017) in the context of machine translation. It is mainly composed of attention layers in both the encoder and the decoder module. The transformer networks introduced the so-called multi-head attention, consisting of several attention layers running in parallel. This multi-head attention allows the model to concurrently access information from different representations of the input vector. Inspired by this work, we designed our multi-head module to read in parallel into different representations of the document while solely accumulate information into the representation of the question.

# 6 CONCLUSION

In this paper, we propose a novel multi-hop reading model designed for question-answering tasks that explicitly require reasoning capabilities. We have designed our model to gather information sequentially and in parallel from a given set of paragraphs to answer a natural language question. Our neural architecture, uses a sequence of token-level attention mechanisms to extract relevant information from the paragraphs and update a latent representation of the question. Our proposed model achieves competitive results on the HOTPOTQA reasoning task and performs better than the current best *published* approach in terms of both Exact Match and $F_1$ score. In addition, we show that an analysis of the sequential attentions can possibly provide human-interpretable reasoning chains.

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

# A  SEQUENTIAL EVOLUTION OF THE ANSWER SPAN PROBABILITIES

This section includes examples from the HOTPOTQA development set that illustrate the evolution of the probabilities for each word to be part of the predicted span, before the first reformulation module and in the answering module presented in Section 4.5. For each example, we show only the text of the two gold paragraphs. ✶ identifies the supporting facts in these visualizations.

**Question:** What is the population according to the 2007 population census of the city in which the National Archives and Library of Ethiopia is located?
**Answer:** 3,384,569 (2.00)
**Predicted answer:** 3,384,569

\*\*\*\*\*\*\* Before Reformulation \*\*\*\*\*\*\*

**Addis Ababa**
Addis Ababa (Amharic: አዲስ አበባ "Addis Abäba " ] , "new flower"; Oromo: "Finfinne" , ] "Natural Spring(s)") or Addis Abeba (the spelling used by the official Ethiopian Mapping Authority), is the capital and largest city of Ethiopia.
✶ It has a population of 3,384,569 according to the 2007 population census, with annual growth rate of 3.8%. This number has been increased from the originally published 2,738,248 figure and appears to be still largely underestimated.

**National Archives and Library of Ethiopia**
✶ The National Archives and Library of Ethiopia, located in Addis Ababa, is the national library and archives of the country.
The library was inaugurated in 1944 by Emperor Haile Selassie and began service with books donated by the emperor.

\*\*\*\*\*\*\* After Reformulation \*\*\*\*\*\*\*

**Addis Ababa**
Addis Ababa (Amharic: አዲስ አበባ "Addis Abäba " ] , "new flower"; Oromo: "Finfinne" , ] "Natural Spring(s)") or Addis Abeba (the spelling used by the official Ethiopian Mapping Authority), is the capital and largest city of Ethiopia.
✶ It has a population of 3,384,569 according to the 2007 population census, with annual growth rate of 3.8%. This number has been increased from the originally published 2,738,248 figure and appears to be still largely underestimated.

**National Archives and Library of Ethiopia**
✶ The National Archives and Library of Ethiopia, located in Addis Ababa, is the national library and archives of the country.
The library was inaugurated in 1944 by Emperor Haile Selassie and began service with books donated by the emperor.

**Question:** Vince Phillips held a junior welterweight title by an organization recognized by what larger Hall of Fame?
**Answer:** International Boxing Hall of Fame
**Predicted answer:** international boxing hall of fame

\*\*\*\*\*\*\* Before Reformulation \*\*\*\*\*\*\*

**International Boxing Federation**
✶ The International Boxing Federation (IBF) is one of four major organizations recognized by the International Boxing Hall of Fame (IBHOF) which sanction world championship boxing bouts, alongside the World Boxing Association (WBA), World Boxing Council (WBC) and World Boxing Organization (WBO).

**Vince Phillips**
Vincent Edwards "Vince" Phillips (born July 23, 1963) is an American former professional boxer who competed from 1989 to 2007.
✶ He held the IBF junior welterweight title from 1997 to 1999, notably scoring an upset victory against then-undefeated Kostya Tszyu to become champion.

\*\*\*\*\*\*\* After Reformulation \*\*\*\*\*\*\*

**International Boxing Federation**
✶ The International Boxing Federation (IBF) is one of four major organizations recognized by the International Boxing Hall of Fame (IBHOF) which sanction world championship boxing bouts, alongside the World Boxing Association (WBA), World Boxing Council (WBC) and World Boxing Organization (WBO).

**Vince Phillips**
Vincent Edwards "Vince" Phillips (born July 23, 1963) is an American former professional boxer who competed from 1989 to 2007.
✶ He held the IBF junior welterweight title from 1997 to 1999, notably scoring an upset victory against then-undefeated Kostya Tszyu to become champion.

**Question:** What award did the writer of Never Let Me Go novel win in 1989?
**Answer:** Man Booker Prize for Fiction
**Predicted answer:** man booker prize for fiction

\*\*\*\*\*\*\* Before Reformulation \*\*\*\*\*\*\*

**Never Let Me Go (novel)**
✱ Never Let Me Go is a 2005 dystopian science fiction novel by Japanese-born British author Kazuo Ishiguro.
✱ It was shortlisted for the 2005 Booker Prize (an award Ishiguro had previously won in 1989 for "The Remains of the Day"), for the 2006 Arthur C. Clarke Award and for the 2005 National Book Critics Circle Award. "
Time" magazine named it the best novel of 2005 and included the novel in its "TIME 100 Best English-language Novels from 1923 to 2005".
It also received an ALA Alex Award in 2006.
A film adaptation directed by Mark Romanek was released in 2010; a Japanese television drama aired in 2016.

**The Remains of the Day**
✱ The Remains of the Day is a 1989 novel by British writer Kazuo Ishiguro.
✱ The work was awarded the Man Booker Prize for Fiction in 1989.
A film adaptation of the novel, made in 1993 and starring Anthony Hopkins and Emma Thompson, was nominated for eight Academy Awards.

\*\*\*\*\*\*\* After Reformulation \*\*\*\*\*\*\*

**Never Let Me Go (novel)**
✱ Never Let Me Go is a 2005 dystopian science fiction novel by Japanese-born British author Kazuo Ishiguro.
✱ It was shortlisted for the 2005 Booker Prize (an award Ishiguro had previously won in 1989 for "The Remains of the Day"), for the 2006 Arthur C. Clarke Award and for the 2005 National Book Critics Circle Award. "
Time" magazine named it the best novel of 2005 and included the novel in its "TIME 100 Best English-language Novels from 1923 to 2005".
It also received an ALA Alex Award in 2006.
A film adaptation directed by Mark Romanek was released in 2010; a Japanese television drama aired in 2016.

**The Remains of the Day**
✱ The Remains of the Day is a 1989 novel by British writer Kazuo Ishiguro.
✱ The work was awarded the Man Booker Prize for Fiction in 1989.
A film adaptation of the novel, made in 1993 and starring Anthony Hopkins and Emma Thompson, was nominated for eight Academy Awards.

**Question:** Tysons Galleria is located in what county?
**Answer:** Fairfax County
**Predicted answer:** fairfax county

\*\*\*\*\*\*\* Before Reformulation \*\*\*\*\*\*\*

**McLean, Virginia**
✱McLean ( ) is a census-designated place (CDP) in Fairfax County in Northern Virginia.
McLean is home to many diplomats, businessmen, members of Congress, and high-ranking government officials partially due to its proximity to Washington, D.C. and the Central Intelligence Agency.
It is the location of Hickory Hill, the former home of Ethel Kennedy, the widow of Robert F. Kennedy.
It is also the location of Salona, the former home of Light-Horse Harry Lee, the Revolutionary War hero.
The community had an estimated total population of 53,673 in 2015, according to estimates prepared by the United States Census Bureau.
It is located between the Potomac River and the town of Vienna.
McLean is known for its luxury homes and its high-end shopping destinations: the Tysons Corner Center and the Tysons Galleria.
The two McLean zip codes - 22101 and 22102 - are among the most expensive ZIP Codes in Virginia and the United States.

**Tysons Galleria**
✱Tysons Galleria is a three-level super-regional mall owned by General Growth Properties located at 2001 International Drive, McLean, Virginia, in Tysons Corner.
It is the second-largest mall in McLean/Tysons Corner, and one of the largest in the Washington metropolitan area.

\*\*\*\*\*\*\* After Reformulation \*\*\*\*\*\*\*

**McLean, Virginia**
✱McLean ( ) is a census-designated place (CDP) in Fairfax County in Northern Virginia.
McLean is home to many diplomats, businessmen, members of Congress, and high-ranking government officials partially due to its proximity to Washington, D.C. and the Central Intelligence Agency.
It is the location of Hickory Hill, the former home of Ethel Kennedy, the widow of Robert F. Kennedy.
It is also the location of Salona, the former home of Light-Horse Harry Lee, the Revolutionary War hero.
The community had an estimated total population of 53,673 in 2015, according to estimates prepared by the United States Census Bureau.
It is located between the Potomac River and the town of Vienna.
McLean is known for its luxury homes and its high-end shopping destinations: the Tysons Corner Center and the Tysons Galleria.
The two McLean zip codes - 22101 and 22102 - are among the most expensive ZIP Codes in Virginia and the United States.

**Tysons Galleria**
✱Tysons Galleria is a three-level super-regional mall owned by General Growth Properties located at 2001 International Drive, McLean, Virginia, in Tysons Corner.
It is the second-largest mall in McLean/Tysons Corner, and one of the largest in the Washington metropolitan area.

**Question:** John MacGregor, Baron MacGregor of Pulham Market was educated at the University of St Andrews and another university established by what monach?
**Answer:** King George IV
**Predicted answer:** king george iv

\*\*\*\*\*\*\* Before Reformulation \*\*\*\*\*\*\*

**King's College London**
King's College London (informally King's or KCL) is a public research university located in London, United Kingdom, and a founding constituent college of the federal University of London.
✶ King's was established in 1829 by King George IV and the Duke of Wellington, when it received its first royal charter (as a college), and claims to be the fourth oldest university in England.
In 1836, King's became one of the two founding colleges of the University of London.
In the late 20th century, King's grew through a series of mergers, including with Queen Elizabeth College and Chelsea College of Science and Technology (in 1985), the Institute of Psychiatry (in 1997), the United Medical and Dental Schools of Guy's and St Thomas' Hospitals and the Florence Nightingale School of Nursing and Midwifery (in 1998).

**John MacGregor, Baron MacGregor of Pulham Market**
John Roddick Russell MacGregor, Baron MacGregor of Pulham Market, OBE PC FKC (born 14 February 1937), is a politician in the United Kingdom.
✶ He was educated at Merchiston Castle School, then at the University of St Andrews (MA economics and history, 1959) and at King's College London (LLB, 1962).
Prior to the 1979 general election he worked for Hill Samuel, a merchant bank.

\*\*\*\*\*\*\* After Reformulation \*\*\*\*\*\*\*

**King's College London**
King's College London (informally King's or KCL) is a public research university located in London, United Kingdom, and a founding constituent college of the federal University of London.
✶ King's was established in 1829 by King George IV and the Duke of Wellington, when it received its first royal charter (as a college), and claims to be the fourth oldest university in England.
In 1836, King's became one of the two founding colleges of the University of London.
In the late 20th century, King's grew through a series of mergers, including with Queen Elizabeth College and Chelsea College of Science and Technology (in 1985), the Institute of Psychiatry (in 1997), the United Medical and Dental Schools of Guy's and St Thomas' Hospitals and the Florence Nightingale School of Nursing and Midwifery (in 1998).

**John MacGregor, Baron MacGregor of Pulham Market**
John Roddick Russell MacGregor, Baron MacGregor of Pulham Market, OBE PC FKC (born 14 February 1937), is a politician in the United Kingdom.
✶ He was educated at Merchiston Castle School, then at the University of St Andrews (MA economics and history, 1959) and at King's College London (LLB, 1962).
Prior to the 1979 general election he worked for Hill Samuel, a merchant bank.

## B  ERROR ANALYSIS

This section includes examples from the HOTPOTQA development set that illustrate the categories of errors presented in Section 4.5. For each example, we show only the text of the two gold paragraphs. ✶ identifies the supporting facts in these visualizations.

**The model stops at the first hop of required reasoning:**

**Question:** What is one of the stars of The Newcomers known for?
**Answer:** superhero roles as the Marvel Comics
**Predicted answer:** chris evans

**Chris Evans (actor)**
✶ Christopher Robert Evans (born June 13, 1981) is an American actor and filmmaker.
Evans is known for his superhero roles as the Marvel Comics characters Steve Rogers / Captain America in the Marvel Cinematic Universe and Johnny Storm / Human Torch in "Fantastic Four" and .

**The Newcomers (film)**
✶ The Newcomers is a 2000 American family drama film directed by James Allen Bradley and starring Christopher McCoy, Kate Bosworth, Paul Dano and Chris Evans.
Christopher McCoy plays Sam Docherty, a boy who moves to Vermont with his family, hoping to make a fresh start away from the city.
It was filmed in Vermont, and released by Artist View Entertainment and MTI Home Video.

**The model fails at comparing two properties:**

**Question:** Who is younger, Wayne Coyne or Toshiko Koshijima?
**Answer:** Toshiko Koshijima
**Predicted answer:** wayne michael coyne)

**Wayne Coyne**
✱ Wayne Michael Coyne(born January 13, 1961) is an American musician.
He is the lead singer, occasional backing vocalist, guitarist, keyboardist, theremin player and songwriter for the band the Flaming Lips.

**Toshiko Koshijima**
✱ Toshiko Koshijima こしじま としこ , Koshijima Toshiko ,born March 3, 1980 in Kanazawa, Ishikawa) is a Japanese singer.
Along with composer, record producer and DJ Yasutaka Nakata, she is a lead vocalist of the electronica band Capsule, which they formed in 1997 when both were 17.
Their formal debut came in 2001 with the release of the single "Sakura".
Two more singles and their debut album, "High Collar Girl", followed the same year.

**The answer does not match all the requirements of the question:**

**Question:** Approximately how many locations is BJ's Wholesale Club operating in, as of early 2008?
**Answer:** 650 locations
**Predicted answer:** 500

**US Vision**
U.S. Vision, a wholly owned subsidiary of Refac Optical Group, is an international optometric dispensary chain.
The vast majority of these locations are leased spaces in large department stores, such as J.C. Penney, Boscov's, and The Bay.
As of May 8, 2007, 500 locations in 47 states and Canada are in operation, consisting of licensed departments and freestanding stores.
✱ In early 2008, due to an acquisition of BJ's Optical Centers located in many BJ's Wholesale Clubs, that number has grown to approximately 650 locations.
U.S. Vision deals mainly in prescription eyewear, contact lenses, and optometry offices.

**BJ's Wholesale Club**
✱ BJ's Wholesale Club Inc., commonly referred to simply as BJ's, is an American membership-only warehouse club chain operating on the United States East Coast, as well as in the state of Ohio.