# OpenReview forum: "Latent Question Reformulation and Information Accumulation for Multi-Hop Machine Reading"
_ICLR.cc/2020/Conference — Reject_

### Official Review · AnonReviewer3 · 2019-10-12
**Official Blind Review #3**

**Rating:** 8

**Review:**

The authors propose a multi-hop latent question reformulation system that performs well in the question-answering setup. The system achieves best current published result on the HotpotQA dataset.

The system achieves that using question-aware representation of the document.

Question: would it be possible to re-generate, at least in an approximate form, one of the reformulations of the question, using decoding? It seems that the visualization of these intermediate forms would allow to understand the model better.

**Experience Assessment:**

I have published one or two papers in this area.

**Review Assessment: Checking Correctness Of Derivations And Theory:**

I did not assess the derivations or theory.

**Review Assessment: Checking Correctness Of Experiments:**

I did not assess the experiments.

**Review Assessment: Thoroughness In Paper Reading:**

I made a quick assessment of this paper.

---

> ### Author Response · Authors · 2019-11-15
> **Answer to Reviewer 3**
>
> We thank the reviewer 3 for its positive feedback.
> ​
> We cannot explicitly reconstruct an approximate form of the question in the intermediate hop and it can be the objective of future work.
> However, in Section 4.5 and Appendix A we observe the attention of the previous layer that highlights the main parts of the document that are used to perform for this reformulation.

---

### Official Review · AnonReviewer1 · 2019-10-23
**Official Blind Review #1**

**Rating:** 3

**Review:**

This paper proposes to iteratively reformulate questions in the latent space for multi-hop question answering. The reformulation of the question depends on the question-aware representation of the documents.

The authors experiment their model on the HotpotQA dataset and achieve the state of the art performance. But, it's important to experiment with some other datasets. One option could be the Complex WebQuestions [1].

Since T is set to 2 in your experiment, is it possible that your model simply predict the intermediate answers and use it to reformulate the question representation? In Talmor et. al paper [1], the SplitQA splits the questions into two subquestions, and appends the answer of the first subquestion to the second subquestion.

Another related work is PullNet by Sun et. al [2].

[1] Talmor, Alon, and Jonathan Berant. "The web as a knowledge-base for answering complex questions." arXiv preprint arXiv:1803.06643 (2018).
[2] Sun, Haitian, Tania Bedrax-Weiss, and William W. Cohen. "PullNet: Open Domain Question Answering with Iterative Retrieval on Knowledge Bases and Text." arXiv preprint arXiv:1904.09537 (2019).

**Experience Assessment:**

I have published one or two papers in this area.

**Review Assessment: Checking Correctness Of Derivations And Theory:**

N/A

**Review Assessment: Checking Correctness Of Experiments:**

I assessed the sensibility of the experiments.

**Review Assessment: Thoroughness In Paper Reading:**

I read the paper at least twice and used my best judgement in assessing the paper.

---

> ### Author Response · Authors · 2019-11-15
> **Answer to Retriever 1**
>
> * We agree that we should not restrict to this single dataset.
> However, since our main contribution is related to multi-hop and parallel reasoning, this configuration allows us to evaluate these skills in a controlled environment without having to rely on a retriever performance.
> To demonstrate the effectiveness of our model in a realistic open-domain configuration, we added in the paper an experiment on a second dataset, the open-domain version of HotpotQA.
> We describe in Section 4.4 how our approach can be combined with a retriever to perform open-domain question answering and show strong results in this evaluation.
> ​
> * Regarding the remark about predicting only the intermediate answer, as we show in Section 4.5 and Appendix A, our model tends to focus on the intermediate answer after the first hop for bridge reasoning type of questions.
> However, it does not rely on an explicit decomposition of the question.
> The DecompRC model [1] adopts a similar approach as in SlitQA with an explicit decomposition of the question but reports lower results than ours.
>
> For this task, an a priori decomposition of the question does not seem reliable.
> Indeed, such decomposition would assume a known structure of the information in the requested documents, like in a database for example.
> However, there is no reason to find such structure in the free text of Wikipedia.
>
> For instance, to the question: What class of instrument does Apatim Majumdar play?
> ​
> The structure of the requested text is:
> Doc1: Apatim Majumdar play sarod.
> Doc2: The sarod is a stringed instrument.
>
> However, it could have also been:
> Doc1: Apatim Majumdar play a stringed instrument.
>
> This example shows that the processing of the question needs to be conditioned on the document itself.
> Indeed, without conditioning on the document, we cannot guess what will be the structure of the information in the requested documents.
>
> [1] Min, Zhong and Zettlemoyer, Multi-hop Reading Comprehension through Question Decomposition and Rescoring, ACL 2019.

---

### Official Review · AnonReviewer2 · 2019-10-26
**Official Blind Review #2**

**Rating:** 3

**Review:**

This paper proposes a model for multi-hop question answering, specifically in a closed domain setting where the relevant paragraphs are present within a few distractor paragraphs. Their model has the following components — (a) a reader module that reads and collect information from paragraphs, (b) a query reformulation module, that reformulates the query to retrieve the next relevant document required to answer the question. Following the effectiveness of multi-head attention, the reader and reformulation module has K-heads that forms independent reformulations. The k-heads are aggregated via a simple summing and the reformulated query representation is used in the next step of the pre-defined hop.

Specifically, they start by encoding the question and paragraphs by BERT embeddings. Next they concatenate the fixed set of paragraphs (10 in their experiments) and encode it with a recurrent NN (bi-GRU) to produce token-level recurrent representation. This is followed by a reading module which dies document question attention in BiDAF (Seo et al., 2017) style followed by a self attention module. Next, in the query reformulation phase a convolution style filter is passed across the token embeddings of the document to gather / pool information required for querying. The current query representation is added to the new pooled representation to obtain the updated query representation. This is followed by an answering module that runs 4 layers of Bi-GRU and has an outlet for computing loss for each of the following — supporting facts , start , end of the answer span representation and a classifier to determine if its a yes-no question or a span answer. The network is trained via supervision for all the aforementioned 4 outlets.

Empricially, this paper shows gain in the distractor setting for HotpotQA dataset.


Strengths:
Query reformulation is an important strategy for IR, and multi-hop QA and it is nice to see that this model adapts it. Each module of the network seems to be carefully designed and the ablation results and analysis are helpful.

Weaknesses:
1. Related work: One of the major weakness of the paper is the missing related work. There are two well-established paper Das et al (ICLR 2019) — Multi-step Retriever-Reader Interaction for Scalable Open-domain Question Answering and Feldman and Yaniv (ACL 2019) — Multi-Hop Paragraph Retrieval for Open-Domain Question Answering that show query reformulation is effective for multi-hop question answering. Das et al 2019 proposes that the query reformulation module is independent of the reader module as long as it has access to the hidden representation of the reader module. This paper builds off from the original papers by carefully designing the reader and the reformulation modules (which is great!), but never mentions any of the above papers. I think that should be fixed.s
2. The setting considered in this paper is closed-domain where the number of paragraphs to be read are fixed and pre-determined. That is a very unrealisitic setting for a general purpose QA system. The paper should consider testing on the open domain setting of the HotpotQA dataset. The other two papers mentioned above (Das et al 2019, Feldman & Yaniv 2019) both test on open-domain setting. Moreover, it has been also shown recently that the distractor setting can easily be fooled and is not a great benchmark for testing the reasoning capabilities.
3. For a real QA system to be deployed in production setting, the system needs to be fast. I am afraid the model proposed by this paper is very computationally expensive. Apart from encoding the question and paragraphs by BERT, they have multiple Bi-GRU encodings (8 additional). That is going to be computationally intensive and the authors should strongly consider other approaches such as replacing bi-grus with transformers that can encode sequences in parallel, and parameter sharing.

For the above strong weaknesses, I am forced to give a low score to the current paper.

**Experience Assessment:**

I have published one or two papers in this area.

**Review Assessment: Checking Correctness Of Derivations And Theory:**

N/A

**Review Assessment: Checking Correctness Of Experiments:**

I carefully checked the experiments.

**Review Assessment: Thoroughness In Paper Reading:**

I read the paper thoroughly.

---

> ### Author Response · Authors · 2019-11-15
> **Answer to Reviewer 2**
>
> 1) Initially, we considered the problem of retrieval out of the scope of this paper and decided not to discuss the open-domain approaches that focus on the retriever part.
> However, we do agree that these two recent papers are closely related to our approach and go in the same direction by adopting a query reformulation mechanism.
> As we present an experiment on the open-domain setting in the updated version of the paper, we now discuss these two papers in Section 5 - Related Work.
> Similarly to our approach, these two papers leverage the idea of question reformulation to perform multi-hop question answering.
> However, we can notice one structural difference with our approach.
> The authors use this reformulation during the retrieval stage, whereas we use the reformulation process directly for the answer extraction task.
> In fact, these approaches are complementary to ours and can be combined together; we can plan such experiments as possible future work.
> We also add [1] in the comparison of the open-domain experiments, in Table 3.
>
> 2) Regarding the fact that the distractor setting would not be a great benchmark for testing reasoning capabilities, we argue that the multi-hop and, more specifically, the parallel reasoning competencies can hardly be handled only by the retriever.
> So we believe that it is pertinent to integrate them into the answering part of the pipeline.
> That is the reason why we consider the distractor setting of HotpotQA as an informative benchmark.
> Nonetheless, to demonstrate the effectiveness of our model in the open-domain setting, we integrated it into an answering pipeline composed of a document retriever followed by our reading model.
> We show that our model, trained on the distractor setting of HotpotQA that contains only ten paragraphs, can scale to a larger number of input documents.
> It allows us to perform open domain-question answering with a pipeline composed of a high recall retriever followed by our proposed reasoning model.
> As mentioned before, we actually show strong results on the open-domain setting of HotpotQA in Section 4.4 of the paper.
>
> 3) We agree that our approach can be considered as computationally expensive.
> However, we show that with a simple high recall retriever, our approach can achieve strong results while limiting the number of documents to process in parallel by the proposed LQR model.
> Indeed, by using a computationally simpler retriever, the overall pipeline might not be more expensive than recurrent based retrieving approaches that necessitate multiple retrieving calls [1, 2].
> ​
> [1] Feldman et al. , Multi-Hop Paragraph Retrieval for Open-Domain Question Answering, ACL 2019.
> [2] Qi et al., Answering Complex Open-domain QuestionsThrough Iterative Query Generation, EMNLP-IJCNLP 2019.

---

### Author Response · Authors · 2019-11-15
**General Response**

We thank all the three reviewers for their interest and valuable feedbacks regarding our paper.
​
One of the primary concerns of reviewers 1 and 2 deals with the experiments conducted only on the distractor setting of the HotpotQA dataset.
We think that this setting is essential and that a strong open-domain system must have good performance in this restricted configuration.
However, we totally agree that this setting might not be entirely realistic for a complete open-domain system.
To answer this concern, we have added an experiment on the open-domain configuration of the HotpotQA dataset as requested.
We integrated our reading model into a pipeline composed of a document retriever (TF-IDF score) followed by our answering model, as previously suggested in [1,2].
This simple combination of a high recall document retriever followed by our proposed reasoning model shows strong positive results on the open-domain HotpotQA dataset, reported in the table below and in Section 4.4 of the paper.
​
In the updated version of the paper, we also discuss some of the references pointed by the reviewers 1 and 2 in Section 5 – Related Work.


  -----------------------------------------------------------------------------------------------------------
                                                                  Ans                 Sup               Joint
                                                            EM        F_1      EM      F_1      EM     F_1
SemanticRetrievalMRS [Nie et al.]  46.50   58.80   39.90   71.50   26.60   49.2
LQR-net (our)                                   43.00   54.00   30.10   58.90   18.90   39.20
Golden Retriever † [Qi et al.]          37.92   48.58   30.69   64.4     18.04   39.13
CogQA [Ding et al.]                         37.60   49.40    23.10   58.5    12.20   35.3
MUPPET [Feldman et al.]                31.07   40.42   17.00   47.71   11.76   27.62
QFE † [Nishida et al.]                       28.70   38.10   14.20   44.40   8.69     23.1
Baseline Model [Yanget al.]            24.68   34.36   5.28     40.98   2.54    17.73
DecompRC [Min et al.]                    N/A     43.26    N/A     N/A      N/A     N/A
  -----------------------------------------------------------------------------------------------------------

  Table 3. Performance comparison on the development set of in the fullwiki setting.
  We compare our model in terms of Exact Match and F_1 scores against the published models at the time of submission (November15th).
  † indicates that the paper does not report the results on thedevelopment set of the dataset; we display their results on the test set

[1] Chen et al. Reading wikipedia to answer open-domain questions, ACL 2017.
[2] Clark et al. Simple and effective multi-paragraph reading comprehension, ACL 2018.

---

### Decision · Program_Chairs · 2019-12-19

**Decision:**

Reject

**Comment:**

This paper proposes a novel approach, Latent Question Reformulation Network (LQR-net), a multi-hop and parallel attentive network designed for question-answering tasks that require multi-hop reasoning capabilities. Experimental results on the HotPotQA dataset achieve competitive results and outperform the top system in terms of exact match and F1 scores. However, reviewers note the limited setting of the experiments on the unrealistic, closed-domain setting of this dataset and suggested experimenting with other data (such as complex WebQuesitons). Reviewers were also concerned about the scalability of the system due to the significant amount of computations. They also noted several previous studies were not included in the paper. Authors acknowledged and made changes according to these suggestions. They also included experiments only on the open-domain subset of the HotPotQA in their rebuttal, unfortunately the results are not as good as before. Hence, I suggest rejecting this paper.